# Effect of Cooling Method on Formability of Laser Cladding IN718 Alloy

**DOI:** 10.3390/ma14133734

**Published:** 2021-07-03

**Authors:** Jianyu Yang, Xudong Li, Fei Li, Wenxiao Wang, Zhijie Li, Guanchao Li, Hualong Xie

**Affiliations:** 1Department of Mechanical Engineering and Automation, Northeastern University, Shenyang 110819, China; jyyang@mail.neu.edu.cn (J.Y.); 1800308@stu.neu.edu.cn (X.L.); 1800322@stu.neu.edu.cn (W.W.); 1800309@stu.neu.edu.cn (Z.L.); 1800306@stu.neu.edu.cn (G.L.); 2Department of Information Science and Engineering, Shenyang University of Technology, Shenyang 110870, China; lifei@sut.edu.cn

**Keywords:** laser cladding, IN718, CA-FE, forced water cooling, mechanical properties

## Abstract

The finite element model (FE) of temperature field of straight thin-walled samples in laser cladding IN718 was established, and the growth of microstructure was simulated by cellular automata (CA) method through macro-micro coupling (CA-FE). The effects of different cooling conditions on microstructure, hardness, and properties of laser-cladding layer were studied by designing cooling device. The results show that the simulation results are in good agreement with the microstructure of the cladding layer observed by the experiment. With the scanning strategy of reducing laser power layer-by-layer, the addition of water cooling device and the processing condition of 0.7 mm Z-axis lift, excellent thin-walled parts can be obtained. With the increase of cladding layers, the pool volume increases, the temperature value increases, the temperature gradient, cooling rate, solidification rate, *K* value gradually decrease, and eventually tend to be stable, in addition, the hardness shows a fluctuating downward trend. Under the processing conditions of layer-by-layer power reduction and water cooling device, the primary dendrite arm spacing reduced to about 8.3 μm, and the average hardness at the bottom of cladding layer increased from 260 HV to 288 HV. The yield strength and tensile strength of the tensile parts prepared under forced water cooling increased to a certain extent, while the elongation slightly decreased.

## 1. Introduction

IN718 is the most important deformation superalloy [1,2], which is mainly a hard alloy precipitated by Mo and Nb. At 700 °C, it is ductile, of high-strength [3], and corrosion resistant [4]. High oxidation resistance was still maintained at 1000 °C [5]. At the same time, it has stable chemical properties and good welding performance at low temperature, which means no cracking tendency after welding. Therefore, IN718 alloy is widely used in aero engines. Laser cladding IN718 alloy is a rapid solidification process, which provides a high cooling rate, so that excellent cladding layer microstructure can be obtained [6]. However, in the cumulative laser cladding process, the continuous accumulation of heat leads to an increase in the molten pool temperature, and finally the decrease of solidification rate. The decrease of solidification rate will lead to microsegregation of elements, coarsening of microstructure, and precipitation of non-equilibrium phases [7].

In order to solve this problem, many experts have done a lot of work. Antonsson [8] used suspension melting method to make the cooling rate reach 20,000 °C/s, which limitedly inhibited the segregation of niobium in the solidification process of IN718 alloy. Tan et al. [9] improved the coating performance by water-cooled laser cladding of aluminum-based amorphous nanocrystalline composite coating. Zhang [10] added liquid nitrogen cooling in the process of laser cladding IN718 alloy, which effectively inhibited the segregation of niobium. Yang et al. [11] designed and developed a water cooling device. The effects of forced cooling on the morphology, columnar crystal volume fraction, and hardness of the cladding layer were studied. More continuous columnar grains were obtained, which increased the volume fraction of columnar grains in the cladding layer. Zen [12,13] uses liquid nitrogen-forced cooling-assisted laser melting magnesium alloy to obtain thinner melting layer and refined microstructure. Wang [14] introduced the forced cooling process into the remelted Ni60 alloy coating, and found that a certain orientation, dense and fine directional solidification structure was formed at the junction of the coating and the substrate. Zhu [15] used forced cooling-assisted laser cladding of Al63Cu27Zn10 coating, and found that the fine dendrites were developed and the microstructure was closely refined, which effectively improved the friction and wear properties of the coating. Ge [16] introduced liquid nitrogen-forced cooling into laser cladding Al-Si alloy and Si3N4 ceramic composite powder. The results show that the microhardness and corrosion resistance of the cladding layer are greatly improved compared with the substrate. Zhang et al. [17] have shown that the microhardness and wear resistance of the coating can be significantly improved by cryogenic treatment after laser cladding of iron-based coating. Zielińskia et al. [18,19,20] described and demonstrated an original laser surface melting technology for non-ferrous metals in liquid nitrogen. All these researchers pointed out the advantages of liquid nitrogen-assisted cooling in accelerating cooling and solidification rate, because liquid nitrogen-assisted cooling can refine the microstructure. Nie et al. [21] studied the microstructure evolution of Ni-based superalloy during laser cladding using random analysis and finite element method, discussed the segregation of niobium, dendrite nucleation and the formation of Laves phase, and revealed the relationship between microstructure and solidification conditions. Dobrzakii et al. [22] discussed the effect of cooling rate on the solidification of AlSi7Cu2 casting alloy, and the grain size, β precipitation component and secondary dendrite arm spacing decreased with the increase of solidification rate.

At present, there are a few studies on the cooling of IN718 alloy by laser cladding. Some experts proposed forced water cooling device but did not study the laser cladding of IN718 alloy. Some proposed that the laser cladding of IN718 alloy was cooled by liquid nitrogen, but the cooling rate and the solidification rate was very fast, resulting in the formation of dense equiaxed grains in the cladding layer. In this paper, the forced water cooling device is designed to promote the epitaxial structure of laser cladding IN718 alloy and improve the mechanical properties along the dendrite direction. At the same time, the influence of forced water cooling on the microstructure and hardness of IN718 alloy is explored. In addition, the finite element model (FE) of temperature field of laser cladding IN718 alloy is established. Through the macro and micro coupling, the cell automaton (CA) method is used to simulate the growth of microstructure in the macroscopic temperature field of the molten pool.

## 2. Materials and Methods

### 2.1. Materials

Materials used in the experiment include: spherical IN718 alloy powder prepared by plasma rotating electrode process (PREP), rolling IN718 alloy substrate. The average particle size of the powder is 100 μm, as shown in Figure 1a, the substrate size is 100 × 50 × 8 mm. Excessive heat accumulation in the cladding process will reduce the quality of the formed parts. In order to avoid heat accumulation, a cooling device is designed, which is mainly composed of two parts: a radiator made of red copper (Figure 1c) and a water-flow groove made of aluminum alloy (Figure 1d).

### 2.2. Experiment Methods

Before the experiment, the powder should be placed in a drying oven at 150 °C for two hours, and the substrate should be polished to remove the oxide layer and oil pollution is cleaned with anhydrous ethanol. The laser cladding experiment was carried out on the SVW80C-3D composite machining center (Sunlight Technology Co., Ltd., Dalian, China) for adding and reducing materials. The equipment used YLS-2000 high-power fiber laser (IPG, Karlsruhe, Germany), with an output spot diameter of 3 mm, a cladding head collimation focal length of 150 mm, and a focusing focal length of 300 mm. A water cooling device (NEU, Shenyang, China) is added to the cladding process, as shown in Figure 2d. Ice water 0 °C is selected as the coolant, and a water pump (Figure 2a) is used to drive the coolant flow. The water flow regulating valve (Figure 2b) is used to adjust the water flow, and a flow meter (Figure 2c) is used to measure the water flow. The coolant dissipates heat from the cladding layer through the cooling device (Figure 2e), and the whole cooling device is connected through the Green joint.

The substrate temperature is measured by thermocouple during laser cladding. In order to observe the microstructure and measure the microhardness, a series of experiments were carried out. The cladding layer is cut along the vertical and parallel scanning direction respectively. Samples are polished and etched with Kallings reagent (40 mL HCl, 40 mL anhydrous ethanol, 2 g CuCl_2_). An OLYMPUS-OLS4100 confocal laser scanning microscope (CLSM, OLYMPUS, Tokyo, Japan) observed the microstructure. The sample was scanned by electron backscatter diffraction (EBSD) using scanning electron microscope (SEM, Oxford Instruments, Oxford, UK). Image-J software is used to measure and calculate the primary dendrite arm spacing. The microhardness test was performed using the HVS-1000 M Vickers microhardness tester (Ledi, Ningbo, China) with a load of 1000 g and a dwell time of 10 s. From the joint of the cladding layer and the substrate, the height direction was measured ten times at an interval of 0.5 mm, and five times at an interval of 1 mm in the same height direction to get the average value. The measurement position is shown in Figure 1b.

The tensile test was carried out by electronic universal material testing machine (WDW-100E, Panasonic, Osaka, Japan). The maximum load is set to 100 kN, the crosshead speed is set to 2 mm/min, and the temperature is set to 300 K. In order to ensure the statistical significance of test results, under the same process parameters, three groups of tensile samples were measured to obtain the average value. The sample size is shown in Figure 3c, unit (mm) and sample thickness—1 mm (Figure 3b). As shown in Figure 3a, the tensile specimen was cut from the thin-walled specimen were 70 × 3 × 40 mm. The main process parameters are shown in the Table 1.

## 3. Macro-Micro Coupling Model

### 3.1. Finite Element Model of Temperature Field

In the process of laser cladding, the temperature field distribution of substrate and cladding material can be solved by three-dimensional transient heat conduction equation [23].
(1)$$\rho c\frac{\partial T}{\partial t}=\frac{\partial }{\partial x}\left(K_{x}\frac{\partial T}{\partial x}\right)+\frac{\partial }{\partial y}\left(K_{y}\frac{\partial T}{\partial y}\right)+\frac{\partial }{\partial z}\left(K_{x}\frac{\partial T}{\partial z}\right)+q$$
where *ρ* is density (kg/m^3^), *c* is specific heat (J/kg·°C), *K* is thermal conductivity (W/m·°C), *q* is the heat generated inside the molten pool (W/m^2^).

The initial conditions can be defined as follows [24]:

For the substrate material:(2)$$T\left(x,y,z,0\right)=T_{0}$$

For all materials:(3)$$T\left(x,y,z,\infty \right)=T_{0}$$

The boundary conditions of powder and substrate are as follow:(4)$$\lambda \frac{\partial T}{\partial n}-{\dot{q}}_{s}+h\left(T-T_{0}\right)+\sigma \varepsilon \left(T^{4}-T_{0}{}^{4}\right)=0$$
where *n* is the surface vertical vector, ${\dot{q}}_{s}$ is the rate of heat input, h is the heat transfer coefficient, σ is the Stefan-Boltzmann constant, ε is the emissivity.

The finite element mesh is shown in Figure 4. The finite element size of the cladding layer is 2.2 mm in width, 0.7 mm in single layer height, and 14 layers are stacked vertically. The grid division of the seventh layer is shown on the left, and the grid division of the fourteenth layer is shown on the right.

### 3.2. Dendritic Growth Model

#### 3.2.1. Macro-Micro Coupling Model

The temperature field of laser cladding was obtained by finite element analysis. The local grid was selected as the interpolation domain, and the temperature value of micro cells was obtained by bilinear interpolation [25], as shown in Figure 5.

#### 3.2.2. Solute Diffusion

The solute diffusion equation is [22]
(5)$$\frac{\partial C_{i}}{\partial t}=\frac{\partial }{\partial x}\left(D_{i}\frac{\partial C_{i}}{\partial x}\right)+\frac{\partial }{\partial y}\left(D_{i}\frac{\partial C_{i}}{\partial y}\right)$$
where *C_S_*, *C_L_* are the solid and liquid solute concentrations. *D_L_*, *D_S_* are the liquid and solid solute diffusion coefficients, respectively (*i = S*, *L*).

#### 3.2.3. Eliminating Grid Heterogeneity

Although the cellular automata method is very effective in simulating dendrite growth, it is still affected by grid anisotropy. In order to reduce the influence of the orthogonally distributed grid on the growth direction of the dendrite, the modified eccentric square algorithm is introduced to simulate the growth of dendrite at random angle [26], which is also consistent with the fact that the dendrite grows freely in the actual solidification process. As shown in Figure 6, the core idea of the algorithm is that with the solidification, the cell A begins to nucleate, and its solid fraction changes. At the same time, a virtual square is generated in the center of the cell A, and its side length *l*(*t*) is always twice the solid fraction *f_s_* of the cell A.
(6)$$l\left(t\right)=2\times f_{s}$$

When the vertices of the virtual square contact with Neumann neighbor cell B, it is captured, and the cell B state changes (if the neighbor cell is a liquid cell, it is transformed into an interface cell, and if it is not a liquid cell, the neighbor cell state does not change). The solid fraction of neighbor cell B changes. A virtual square is generated in the neighbor cell B, and the center of the virtual square in cell B is the vertex of the virtual square in cell A when the virtual square is the largest in cell A, which is expressed as:(7)$$\left\{ \begin{array}{l} x_{B}=x_{A}+\sqrt{2}f_{s}cos\theta \\ y_{B}=y_{A}+\sqrt{2}f_{s}sin\theta \end{array}\right.$$

The surrounding cells are captured in turn and the state changes. When there is no liquid phase in the Neumann adjacent cells of the interface cell, the interface cell changes into a solid cell until the solidification is completed.

## 4. Results and Discussion

### 4.1. Results of Finite Element Temperature Field

Process parameters: laser power 1200 W, scanning rate 10 mm/s, powder feeding rate 1.4 RPM, Z-axis lift 0.7 mm, with constant laser power vertically stacked 14 layers, the simulation results are shown in Figure 7. On the left side of Figure 7, it can be found that the temperature distribution of the cladding layer is ellipsoidal from the three-view diagram. The temperature is set to the solid line temperature of 1260 °C, and the temperature distribution of the molten pool is ellipsoidal on the right side of Figure 7. The isothermal lines at the front of the molten pool are dense and the temperature gradient is large. The isothermal lines at the rear of the molten pool are sparse and the temperature gradient is small, which is consistent with the actual molten pool morphology.

As shown in Figure 8a, the temperature data of nodes 76,169 and 76,378 are obtained by finite element calculation in the simulation model. The simulated cladding layer is 35 mm in length, 2.2 mm in width, 0.7 mm in single layer height, and 14 layers are stacked vertically. As shown in Figure 8b, the temperature data of points A and B are collected in the experiment, and the position of the point corresponds to the position of the simulation node. The experimental cladding layer is 35 mm in length, 2.5 mm in width, 0.7 mm in Z-axis lift, and 14 layers are printed vertically. As shown in Figure 9, it can be found that the simulation nodes 76,169 and 76,378 temperatures have similar variation trends with the experimental measured temperatures of A and B. At the beginning of laser cladding, the temperature rises sharply to the maximum temperature of about 500 °C, and then the temperature slowly decreases and tends to be stable. The temperature fluctuates between 220 °C and 400 °C until the end of cladding. The simulation results are in good agreement with the experimental results.

In the cladding process, with the increase of the cladding layer, the continuous accumulation of heat causes the volume of the molten pool to become larger. When the heat accumulates too much, it will lead to the collapse of the cladding layer and affect the forming effect. Therefore, a layer-by-layer scanning path to reduce the laser power is proposed to reduce the heat accumulation, Figure 10a shows a uninterrupted scanning strategy with constant laser power, the main process: turn on the laser, scanning length 35 mm, turn off the laser, lift 0.7 mm along the Z axis, scanning cycle. Figure 10b shows an intermittent scanning strategy with reducing power layer by layer, the main process: turn on the laser, scanning length 35 mm, turn off the laser, scanning 10 mm, lift 0.7 mm along the Z axis, reduce the laser power, scanning cycle. The numerical variation of power reduction layer-by-layer is shown in Table 2.

Through the finite element calculation, the simulation results are analyzed. The temperature distribution range of the molten pool is obtained by setting the solid phase line at 1260 °C, and the change of the molten pool morphology is analyzed layer by layer. As shown in Figure 11, it can be found that at the beginning of the cladding, the input heat continues to accumulate, and the molten pool volume gradually increases. With the increase of the number of cladding layers, the input and output of the cladding heat reach equilibrium, and the molten pool volume tends to be stable. According to the two scanning paths, the temperature distribution results of the molten pool from the first layer to the tenth layer are compared. As shown in Figure 11, the molten pool with constant laser power is on the left, and the molten pool with layer-by-layer reduction of laser power is on the right. Compared with the morphological changes of the molten pool on the left, the volume of the molten pool on the right is reduced, and the molten pool is more stable, so as to ensure the high quality of the part forming. As the molten pool volume decreases, the depth of cladding layer decreases, and the molten pool becomes shallow. The shallower molten pool will improve the ability to obtain columnar crystals, which will delay the transformation of columnar grains to equiaxed grains (CET) during solidification.

As shown in Figure 12, by comparing the cross-sectional images of processing methods (a) and (b), the processing effect of processing method (a) is as follows: the width of the bottom cladding layer is narrow. With the increase of the number of cladding layers, the cladding layer becomes wider, and the overall forming thickness is uneven. The overall forming thickness of machining mode (b) is uniform. Compared with the above simulation results in Figure 11, the finite element model is in good agreement with the experimental results.

By observing the cross-sectional images of processing methods (b), (c), and (d), it can be found that with the decrease of Z-axis lift amount, the width of cladding layer becomes wider and the cumulative efficiency decreases. When the Z-axis lift amount is 0.7 mm, thin-walled parts with narrow cladding layer thickness, high cumulative efficiency, and uniform forming thickness can be obtained.

By observing the cross-sectional images of processing methods (b), (e), and (f), it can be found that the cross-sectional changes of cladding layers in (e) and (f) are not uniform, which may be mainly due to the influence of slow water cooling and air cooling on the accumulation of heat, resulting in the instability of heat in molten pool and affecting the uniformity of cladding thickness.

In the finite element simulation, the temperature field of 14 cladding layers was simulated. There were 35 load steps in each cladding layer, the maximum temperature T and temperature gradient *G* of the molten pool were recorded in five load steps with spacing, and the average value was solved.
(8)$$\left\{ \begin{array}{l} K=G^{3.4}/R\\ V=G\times R\\ V=\partial T/\partial t \end{array}\right.$$

The cooling rate *V*, solidification rate *R* and *K* were solved by Formula (8), and the analysis results are shown in Figure 13. It can be found that with the increase of cladding layer, the temperature value increases, the temperature gradient, cooling rate, solidification rate, and *K* value decrease, and finally tend to be stable. The main reason is that at the beginning of the cladding, the substrate is at room temperature, the heat dissipation is fast, the cladding temperature is low, and the *K* value, etc., is large. With the increase of the cladding layer, the cladding layer is used as the substrate, and the input of heat is greater than the output, resulting in the increase of temperature and the decrease of *K* value, etc. When the input and output of heat are balanced, these values tend to be stable. Compared with the four simulation conditions, the power reduction and water cooling processing conditions effectively reduce the temperature of the cladding layer and improve the *K* value, etc., of the cladding layer.

During the solidification process, when the solidification of the first layer occurs, the rolling IN718 alloy is used as the substrate, and the solidified IN718 alloy powder is used as the substrate from the second layer. Because the cladding is carried out in the air and the laser provides a high temperature environment, it is inevitable to react in the molten pool and form impurities. The impurities introduced in the molten pool change the heterogeneous nucleation in the solidification process of the molten pool, and then affect the dendrite growth in the molten pool. The microstructure of the cladding layer changes after the solidification, which affects the performance of the formed parts.

With the increase of cladding layer, the relationship between heterogeneous nucleation number and cladding layer number can be approximately expressed by the following expression [28]:(9)$$N∼n\left(Ma+A\right)+N_{0}$$
where *N* is proportional to *Ma* and *A*, *N* represents the nucleation number, *Ma* is the Marangoni number, *A* is the number of impurity elements introduced, *N*_0_ represents the original nucleation number, *n* represents the number of the cladding layer.
(10)$$Ma=\frac{d\gamma }{dT}\frac{dT}{dx}\frac{L^{2}}{\eta a}$$
where $\frac{d\gamma }{dT}$—thermal gradient of surface tension, $\frac{dT}{dx}$—temperature gradient, η—dynamic viscosity, a—thermal diffusivity, L—characteristic scale.
(11)$$\frac{G^{n}}{R}=K<K_{CET}=a\times {\left[\sqrt[{3}]{\frac{-4\pi N}{3ln\left(1-\varphi \right)}}\times \frac{1}{n+1}\right]}^{n}$$
where *a*, *n* are the constants related to the material, the values are *a* = 1.25 × 10^6^, *n* = 3.4 [29], *φ* is the volume fraction of equiaxed grain.

With the increase of the cladding layer, the nucleation number in the molten pool will continue to increase, according to the CET model, the equation of the original stability condition is no longer valid. With the increase of the critical value of CET, when the ratio of temperature gradient to solidification rate does not reach the critical value of CET, the epitaxial growth of columnar crystals will be interrupted. In general, with the increase of cladding layer, the temperature gradient decreases, the solidification rate decreases, resulting in the decrease of K value, but the critical value of CET increases, which leads to the promotion of CET, that is, the position of CET is advanced, or even CET does not occur, and then the microstructure of all equiaxed grains is formed.

As shown in Figure 14, it can be found that with the increase of nucleation number, in order to ensure the epitaxial growth of columnar grains at the same solidification rate, higher temperature gradient must be required, so that the epitaxial growth conditions of columnar grains are higher, and even the equiaxed grain morphology is inevitable.

### 4.2. Simulation Results of CA

As shown in Figure 15, by comparing the simulated dendrite morphology with the experimental observation, it can be found that the simulation can better simulate the dendrite orientation. The simulation selects the temperature field region between the cladding layers as the interpolation domain, and simulates the micro dendrite morphology of the adjacent two cladding layers. It is found that the columnar grains are mixed with equiaxed grains, and the growth direction of columnar grains changes in “Z” shape, which conforms to the temperature gradient direction of the above round-trip scanning path.

### 4.3. Microstructure Analysis of Cladding Layer

The flow rate is 7.5 L/min. The laser power was reduced layer-by-layer to accumulate fourteen layers vertically. As shown in Figure 16, Figure 16a is the substrate of cast IN718 alloy, and its microstructure is shown as massive equiaxed grains. Figure 16b is the bottom of the cladding layer combined with the substrate. The microstructure is mainly shown as columnar dendrites with a large angle between the growth direction and the vertical direction, and there is also a small amount of equiaxed grains. This is mainly because the junction of the cladding layer and the substrate is ellipsoid, and the temperature gradient along the normal direction of the junction is the largest, and the dendrites grow preferentially along the direction of the temperature gradient. Combining the first cladding layer with the substrate, the contact volume of the molten pool is large, the heat dissipation is fast, the heat dissipation is around the semi-ellipse, the temperature gradient is large, the cooling rate is fast, the dendrite growth is relatively dense, and the deviation between the dendrite growth angle and the vertical direction is large. Figure 16c is the middle of the cladding layer. The microstructure is mainly shown as the columnar dendrite structure with small deviation between the growth direction and the vertical direction. It is mainly because the former cladding layer is used as the substrate in the middle of the cladding layer, which leads to less contact volume of the cladding layer, relatively consistent heat dissipation direction, and relatively consistent temperature gradient direction. With the increase of cladding layer, the temperature gradient decreases, but the gradient value is still large, and the ratio of temperature gradient and solidification rate still meet the requirements of columnar grain growth. At the same time, each layer of remelting melts the top of the cladding layer of the previous layer, eliminates most of the equiaxed grains, and a small amount of equiaxed crystal remains in the bonding area of the cladding layer. Figure 16d shows the top of the cladding layer, and the microstructure is shown as CET. Complete equiaxed grain morphology appears at the top of the cladding layer, mainly because the temperature gradient at the top of the cladding layer decreases.

Observation of longitudinal section microstructure of cladding layer [30] is shown in Figure 17. Figure 17a is the cross section of the cladding layer, it can be found that the junction between the cladding layers shows the morphology of the middle bulge, and with the increase of the cladding layer, the height of the middle bulge increases, which leads to the decrease of the flow of the liquid molten pool to both sides, so as to ensure the thickness of the formed part is consistent. Figure 17b is the longitudinal section of the cladding layer, it can be found that the junction between the cladding layers is generally stable and accompanied by local fluctuations. The local fluctuation between the cladding layers indicates that the current molten pool state is unstable. Once the liquid molten pool is unstable, this state will accumulate until the end of the whole cladding process, resulting in uneven morphology at the top of the longitudinal section of the cladding layer and poor quality of the formed parts. Figure 17c is the microstructure at the bottom of the longitudinal section of the cladding layer. It can be found that there are still equiaxed grains between the cladding layers, indicating that the equiaxed grains are not completely eliminated during the remelting process. Figure 17e is the local amplification figure of Figure 17c. It can be found that there are columnar grains with different growth directions at the bottom, and the formation is dense, mainly due to the large cooling rate at the bottom. Figure 17d is the microstructure of the top of the longitudinal section of the cladding layer. It can be found that CET occurs obviously at the top of the cladding layer, and the top is completely equiaxed grain growth. Figure 17f is the local amplification figure of Figure 17d. It can be found that the upper part of the cladding layer is mainly columnar dendrite morphology, and the forming is relatively sparse.

### 4.4. Effect of Cooling Method on Formability

As shown in Figure 18a, the cladding depth with water cooling device is larger, and the growth height of the columnar dendrite is larger. Figure 18c, d corresponds to the local microstructure of Figure 18a,b, respectively. Observation of EBSD shows that the lower part is the growth of columnar dendrite, and the upper part is the growth of block equiaxed grains.

As shown in Figure 19, the middle of the cladding layer was selected as the columnar dendrite observation area, and the primary dendrite arm spacing (PDS) was measured by Image-J software. Primary dendrite arm spacing is measured by dividing the distance between dendrite gaps by the number of dendrites. The average value was obtained by multiple measurements, and multiple regions were selected for repeated operation to measure the primary dendrite arm spacing. The results are shown in Figure 20. It can be found that the primary dendrite arm spacing decreases with the increase of cooling rate. Compared with the four processing conditions, the maximum cooling rate is provided by reducing the power water cooling (7.5 L/min) condition, and the primary dendrite arm spacing can reach about 8.3 μm. The minimum cooling rate is provided by constant laser power air cooling, and the primary dendrite arm spacing reaches about 11.02 μm.

The Vickers hardness trend of the cladding layer under the four conditions is shown in Figure 21a. It can be found that the hardness fluctuates and decreases with the increase of cladding layer. The hardness at the bottom of the cladding layer is higher than that at the middle, and the hardness at the top is the smallest, which may be due to the strengthening effect of the precipitates $\gamma^{\prime \prime }$ phase on the microstructure. It can be seen that the average hardness of the cladding layer under the processing condition of reducing the laser power water cooling (7.5 L/min) is the highest, and the average hardness of the cladding layer under the condition of constant laser power air cooling is the lowest, which may be that the increase of the cooling rate reduces the content of the brittle and hard phase Laves phase, and then more strengthening phase $\gamma^{\prime }$, $\gamma^{\prime \prime }$ phase precipitates, resulting in improving the hardness. The highest hardness at the bottom of the two processing conditions are 288 HV and 260 HV, and the highest hardness at the top are 253 HV and 236 HV, respectively. It can be concluded that water cooling has a certain effect on improving the hardness, but the degree is limited.

Tensile parts are made by using constant laser power under air cooling and water cooling conditions (7.5 L/min) respectively, and tensile tests are carried out [31]. The results of the tensile curve are shown in Figure 21b. The yield strength, tensile strength, and elongation of the tensile samples were compared, as shown in Figure 21c. Tensile data are shown in the Table 3. It can be found that the yield strength and tensile strength of the tensile samples prepared under the condition of water cooling are increased to a certain extent, while the elongation is slightly decreased, which may be mainly because the cooling rate of the cladding process is increased by water cooling, resulting in the formation of more compact dendrite morphology, and then improving the mechanical properties of the tensile samples.

## 5. Conclusions

The finite element model (FE) of the temperature field of the straight thin-walled parts formed by laser cladding was established. Through the macro-micro coupling (CA-FE), the growth of the microstructure of the laser cladding IN718 alloy was simulated by the cellular automata (CA) method. It was found from the simulation results that the temperature simulation results of the finite element temperature field model were consistent with the experimental temperature measurement results, and the simulation results of the microstructure were consistent with the experimental observation results. The effects of different cooling conditions on microstructure, hardness, and properties of cladding layer were studied by designing cooling device.

With the increase of cladding layer, the volume of molten pool becomes larger and finally tends to be stable. The processing conditions of the scanning strategy of reducing laser power layer-by-layer, additional water cooling device, and 0.7 mm of Z-axis lift are adopted to avoid the accumulation of heat, so as to obtain thin-walled parts with good forming quality.With the increase of cladding layer, the heat input is greater than the output, and the temperature increases. The temperature gradient G, cooling rate V, solidification rate R and K value gradually decrease. When the heat input and output are balanced, these parameters tend to be stable. Under the condition of scanning strategy of layer-by-layer reducing laser power and adding water cooling device, the cladding temperature is effectively reduced, and the temperature gradient, cooling rate, solidification rate, and K value are improved.Under the condition of forced water cooling, the growth of columnar dendrites was promoted. The primary dendrite arm spacing decreased with the increase of cooling rate. Reducing the power water cooling condition provided the maximum cooling rate, and the primary dendrite arm spacing could reach about 8.3 μm.The hardness showed a fluctuating downward trend with the increase of the cladding layer. The average hardness at the bottom of cladding layer increased from 260HV to 288HV under the processing conditions of reducing the laser power water cooling 7.5 L/min.The yield strength and tensile strength of the tensile specimens prepared under forced water cooling conditions were improved to a certain extent, but the elongation decreased slightly.

## Figures and Tables

**Figure 1 materials-14-03734-f001:**
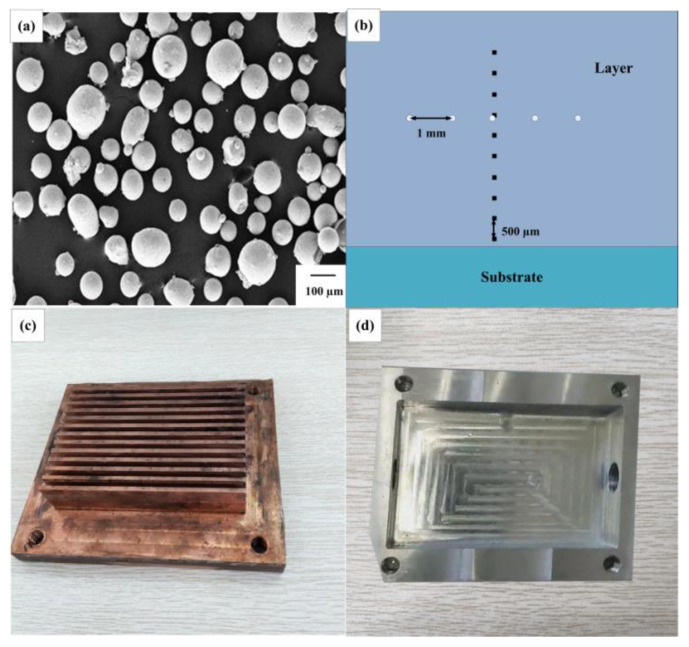
Experimental materials: (**a**) morphology of IN718 alloy powder (200×), (**b**) point location of hardness test; cooling device: (**c**) radiator, (**d**) water-flow groove.

**Figure 2 materials-14-03734-f002:**
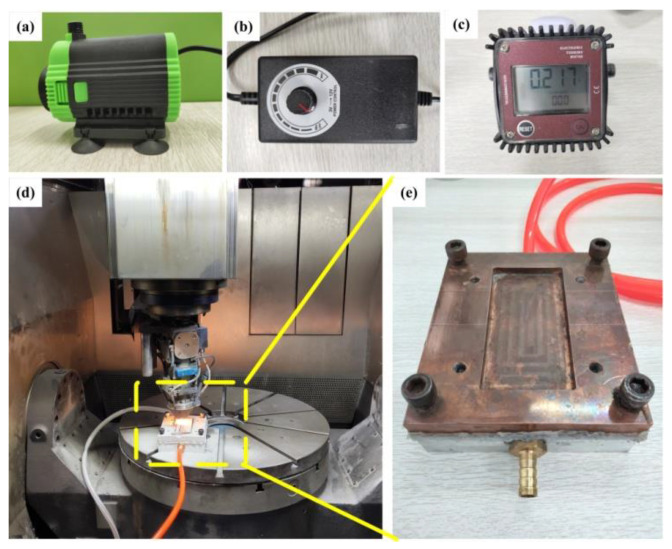
Laser cladding with additional cooling device: (**a**) water pump, (**b**) water flow regulating valve, (**c**) flow meter, (**d**) laser cladding process with additional cooling device, (**e**) cooling device.

**Figure 3 materials-14-03734-f003:**
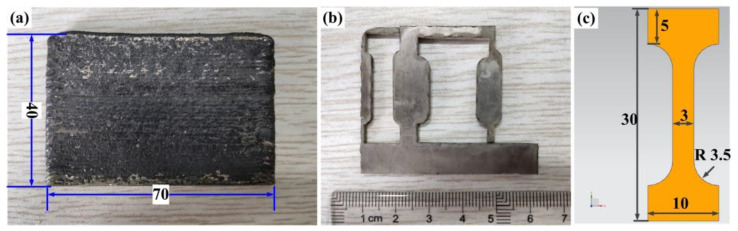
Preparation of tensile sample: (**a**) straight thin-walled sample, (**b**) cutting positions of the tensile sample, (**c**) size of the tensile sample.

**Figure 4 materials-14-03734-f004:**
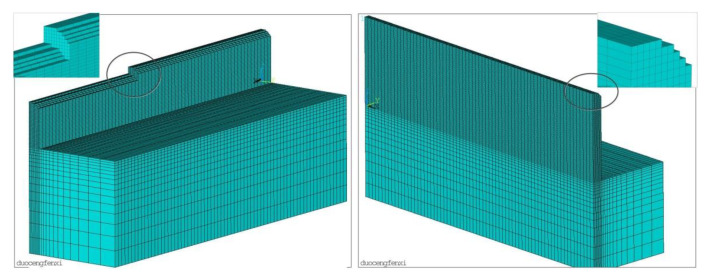
Finite element meshing model.

**Figure 5 materials-14-03734-f005:**
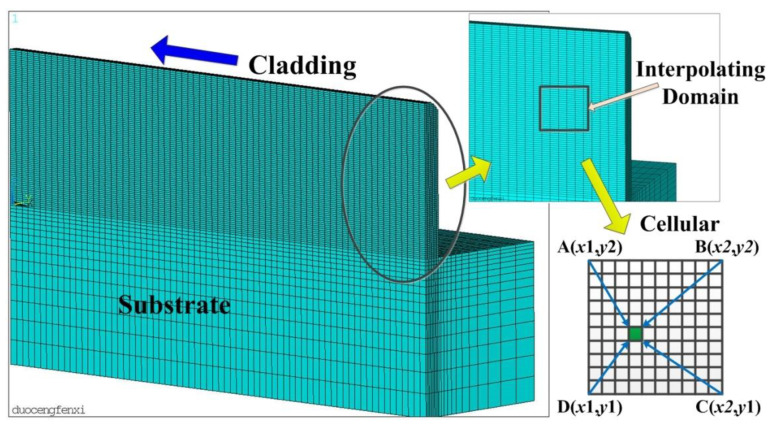
Macro-micro coupling model.

**Figure 6 materials-14-03734-f006:**
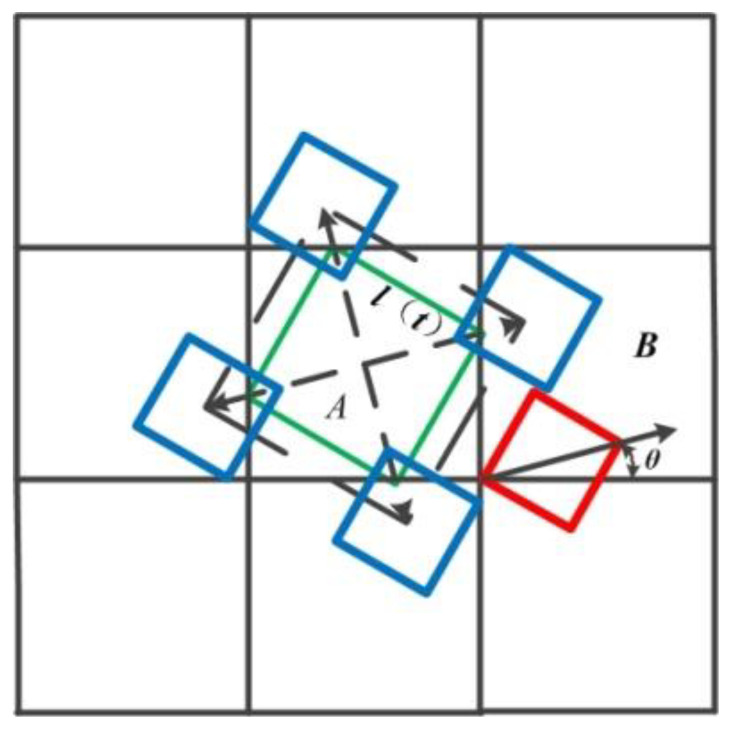
Diagram of modified eccentric square algorithm [27].

**Figure 7 materials-14-03734-f007:**
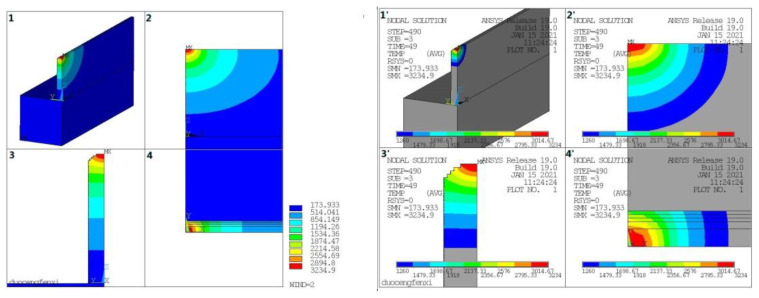
Simulation results of laser cladding temperature field and molten pool temperature field distribution: 1—three-dimensional diagram, 2—front view, 3—side view, 4—planform.

**Figure 8 materials-14-03734-f008:**
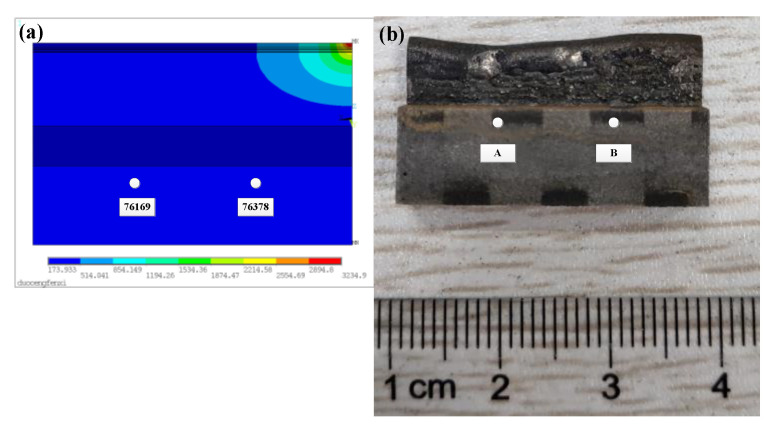
Comparison of observation points: (**a**) simulation, (**b**) experiment.

**Figure 9 materials-14-03734-f009:**
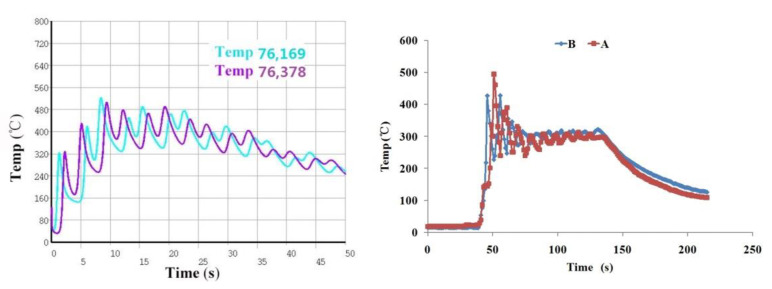
Simulation and experimental temperature comparison.

**Figure 10 materials-14-03734-f010:**
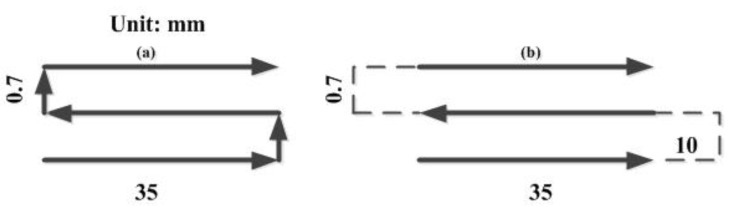
Scanning path comparison: (**a**) constant laser power in continuous scanning, (**b**) reduced laser power layer by layer in discontinuous scanning.

**Figure 11 materials-14-03734-f011:**
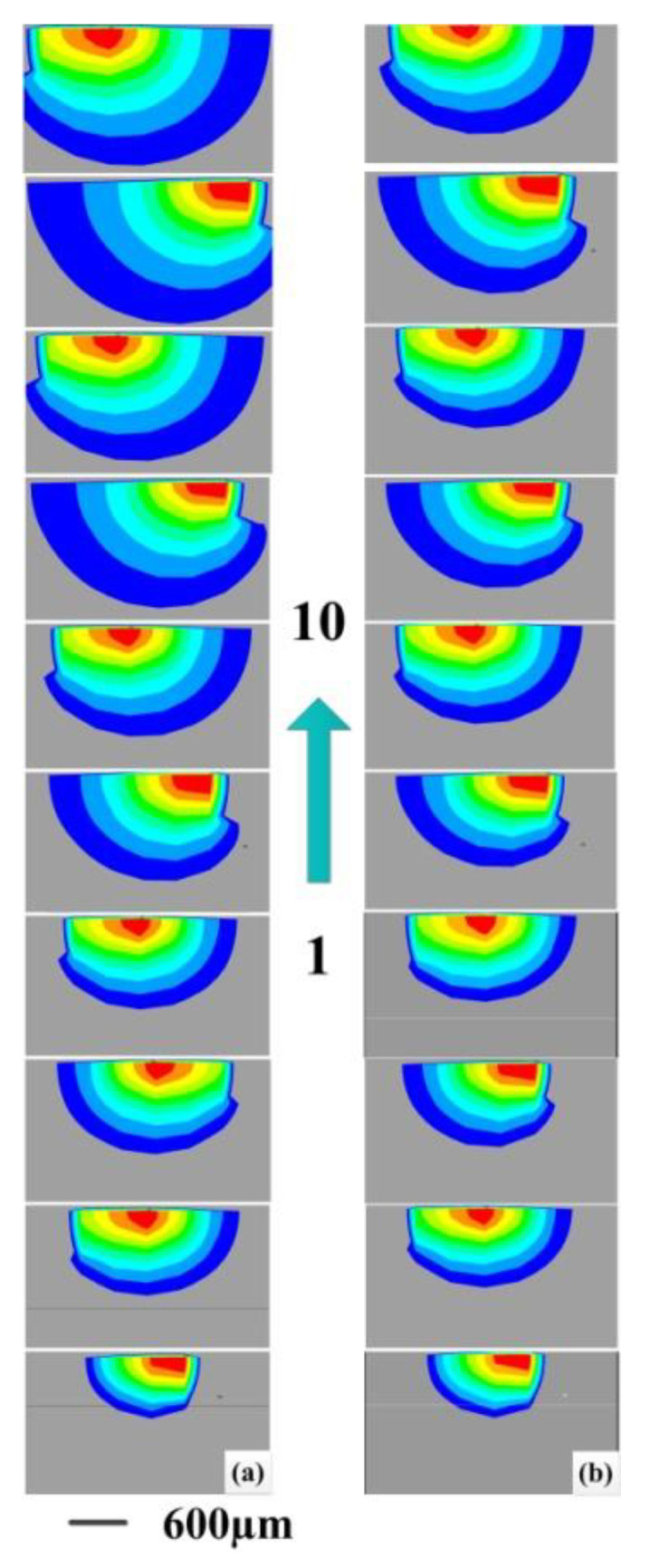
Comparison of multi-layer simulation molten pool with (**a**) constant power and (**b**) layer-by-layer reduced power.

**Figure 12 materials-14-03734-f012:**
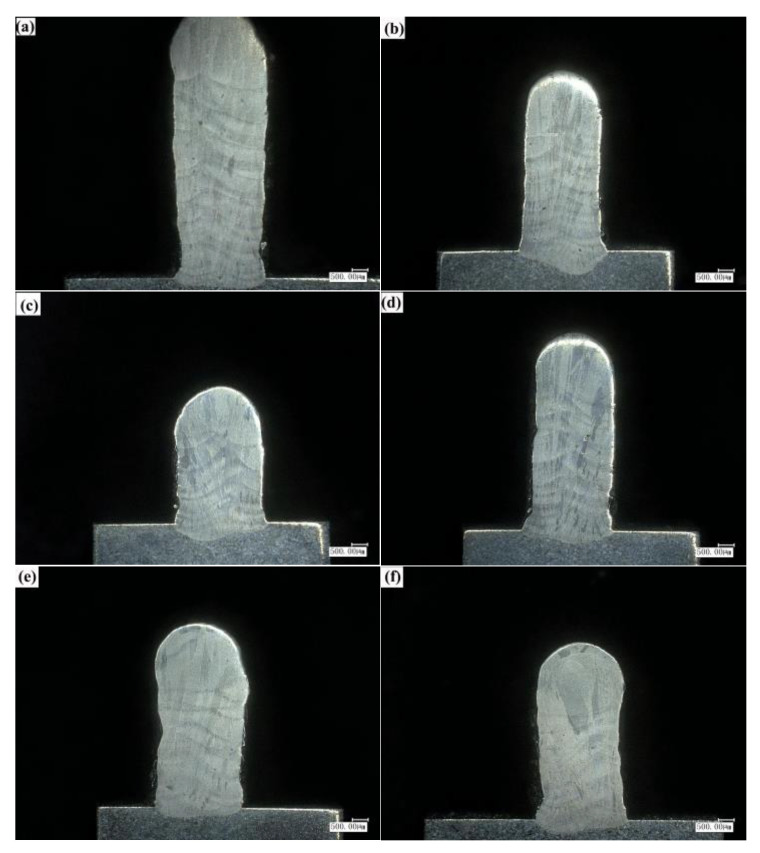
Cross section of cladding layer: (**a**) Constant laser power, air cooling, Z-axis lift 0.7 mm processing conditions, (**b**) decreasing laser power step-by-step, water cooling 7.5 L/min, Z-axis lift 0.7 mm processing conditions, (**c**) decreasing laser power layer-by-layer, water cooling 7.5 L/min, Z-axis lift 0.2 mm processing conditions, (**d**) decreasing laser power layer-by-layer, water cooling 7.5 L/min, Z-axis lift 0.5 mm processing conditions, (**e**) decreasing laser power layer-by-layer, water cooling 4 L/min, Z-axis lift 0.7 mm processing conditions, (**f**) decreasing laser power layer-by-layer, air cooling, Z-axis lift 0.7 mm processing conditions.

**Figure 13 materials-14-03734-f013:**
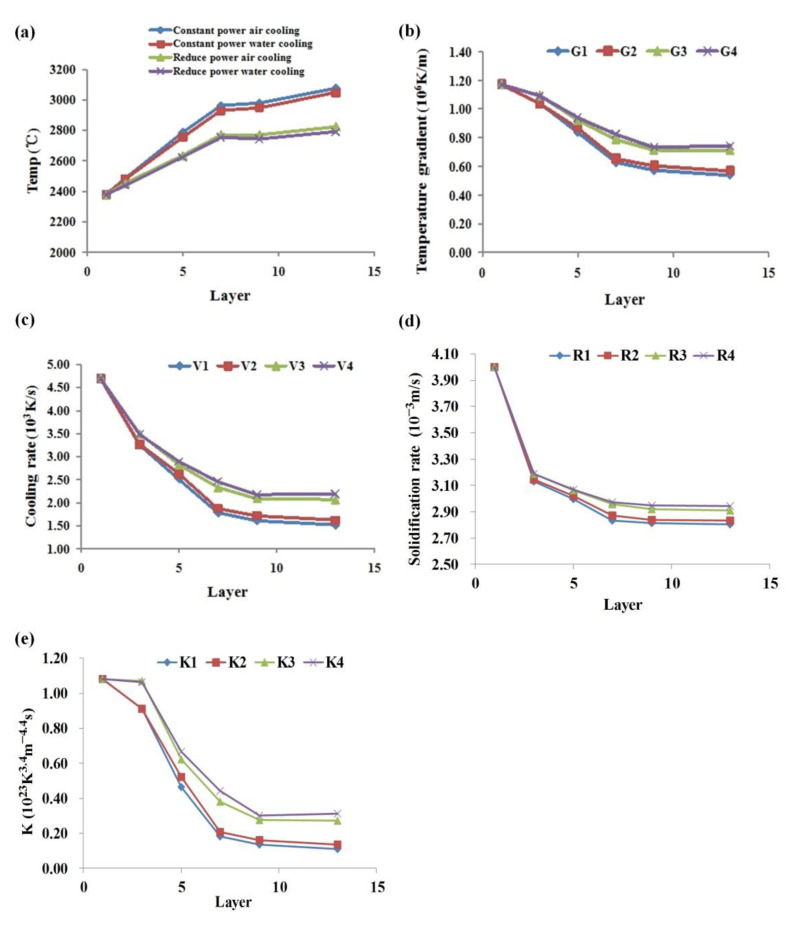
(**a**) Temperature, (**b**) temperature gradient *G*, (**c**) cooling rate *V*, (**d**) solidification rate *R*, (**e**) *K* value variation with cladding layer, 1—constant power air cooling, 2—constant power water cooling, 3—reduce power air cooling, 4—reduce power water cooling.

**Figure 14 materials-14-03734-f014:**
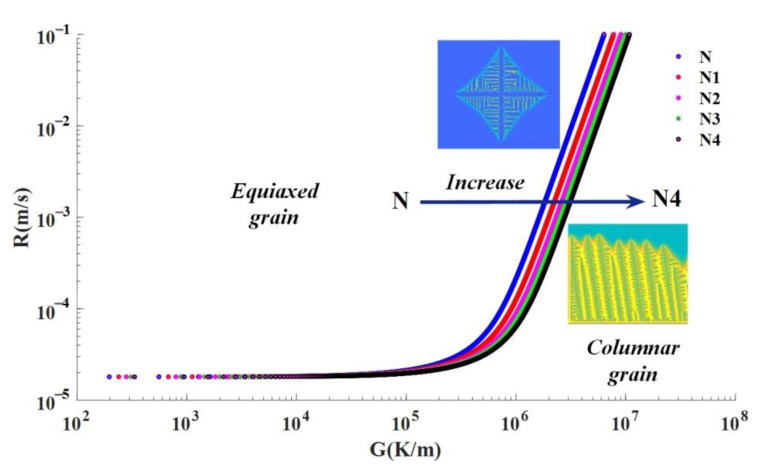
Curves of *K* value varying with nucleation *N*.

**Figure 15 materials-14-03734-f015:**
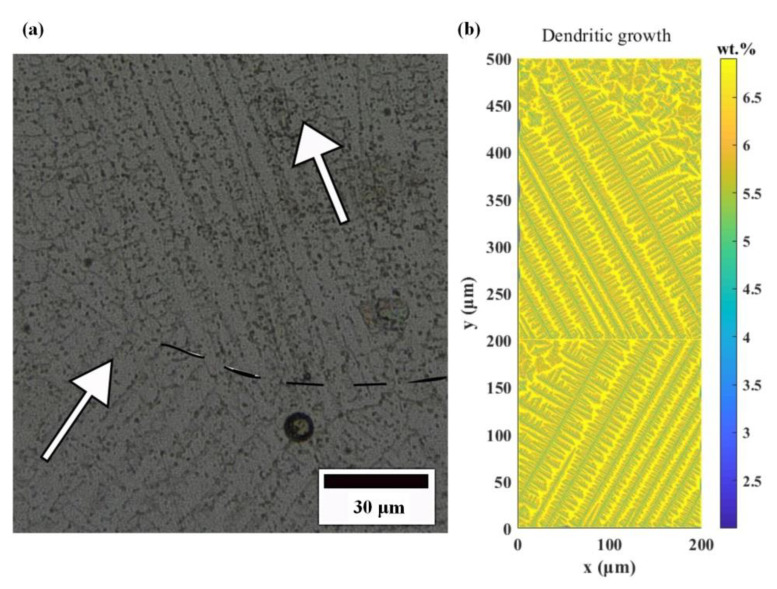
Comparison of dendrites morphology: (**a**) experiment, (**b**) simulation.

**Figure 16 materials-14-03734-f016:**
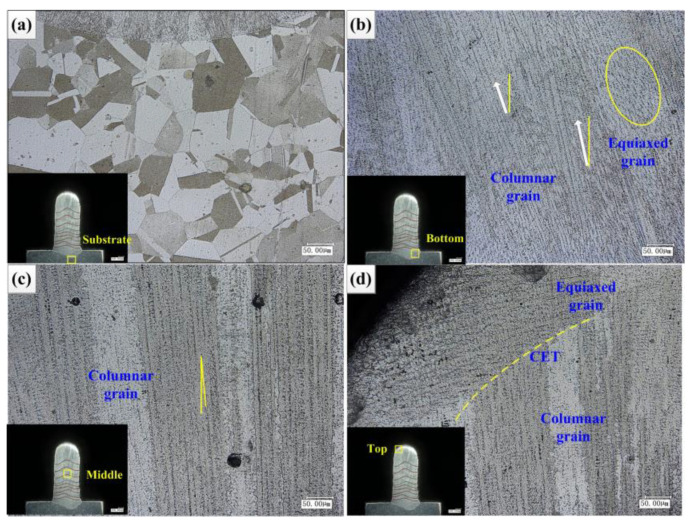
Morphology of cross section of cladding layer: (**a**) substrate, (**b**) bottom, (**c**) middle, (**d**) top.

**Figure 17 materials-14-03734-f017:**
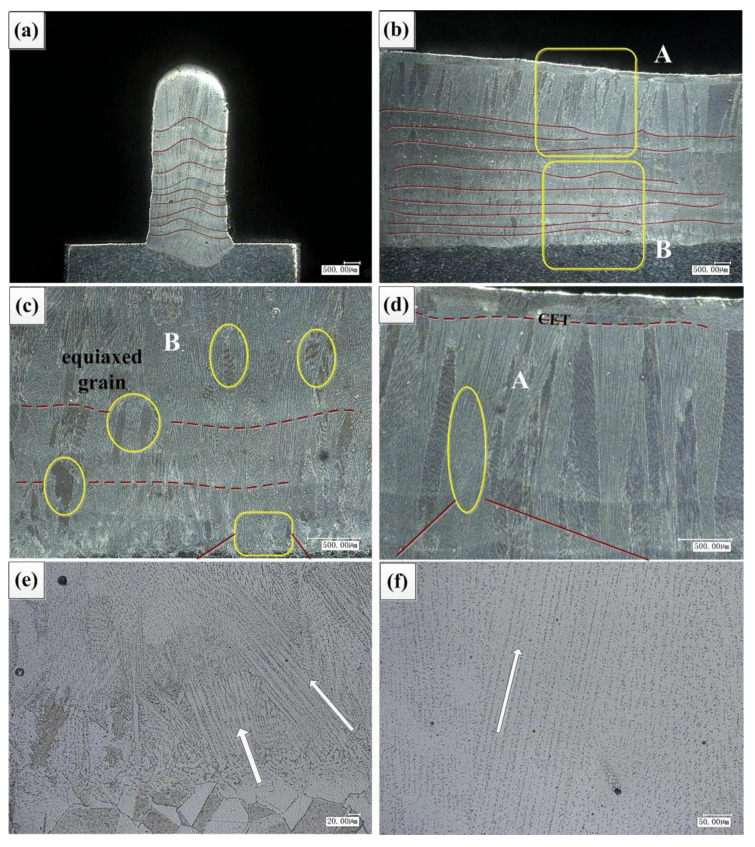
Longitudinal section morphology of cladding layer: (**a**) morphology of cross section, (**b**) longitudinal section morphology, (**c**) B-bottom of longitudinal section, (**d**) A-top of longitudinal section, (**e**) enlargement of bottom, (**f**) enlargement of top.

**Figure 18 materials-14-03734-f018:**
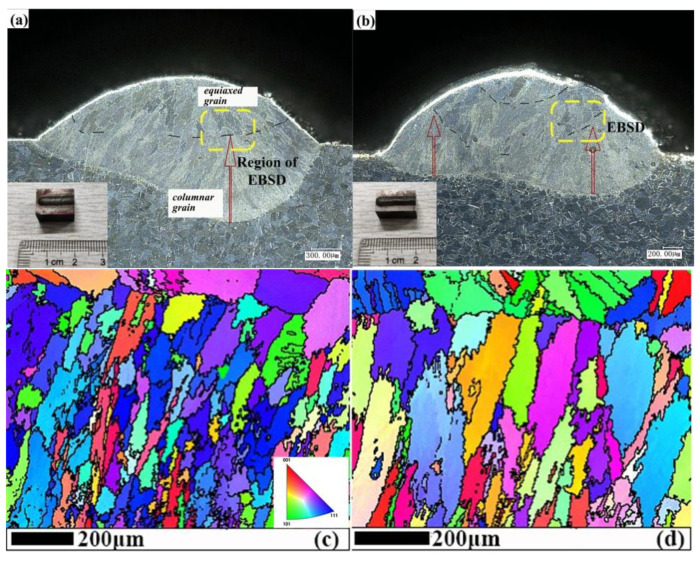
Cross section of cladding layer: (**a**) additional water cooling device (7.5 L/min), (**b**) air cooling, (**c**) EBSD map of (**a**), (**d**) EBSD map of (**b**).

**Figure 19 materials-14-03734-f019:**
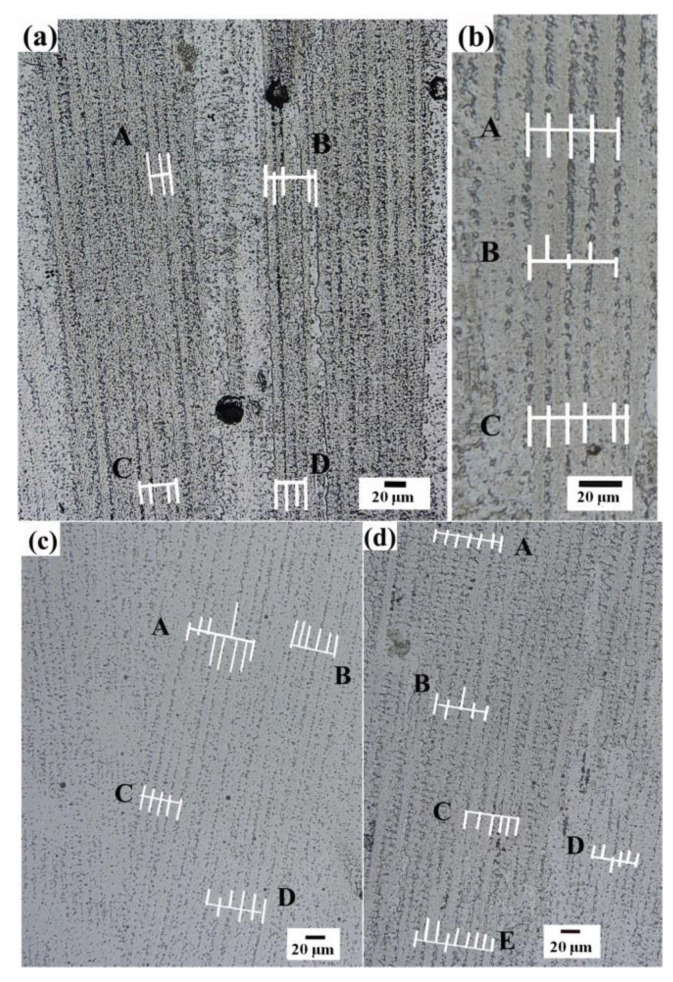
Primary dendrite arm spacing under (**a**) reducing power water cooling (7.5 L/min) condition, (**b**) reducing power air cooling condition, (**c**) reducing power water cooling (4 L/min) condition, (**d**) constant power air cooling condition.

**Figure 20 materials-14-03734-f020:**
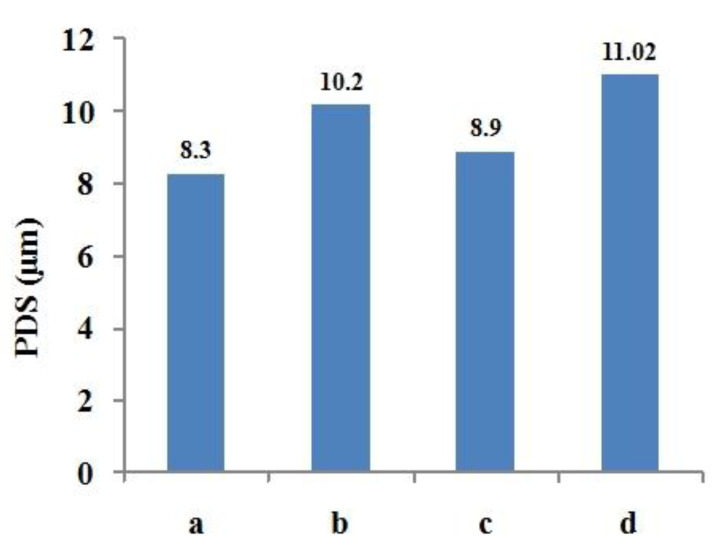
PDS of each processing condition.

**Figure 21 materials-14-03734-f021:**
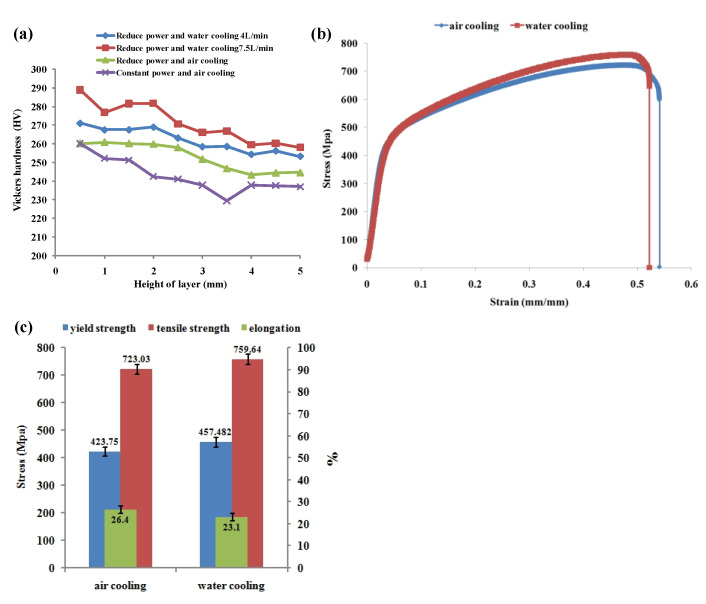
Analysis of the effect of cooling on formability: (**a**) the Vickers hardness trend of the cladding layer under the four conditions, (**b**) the tensile curve, (**c**) the yield strength, tensile strength and elongation of the tensile samples.

**Table 1 materials-14-03734-t001:** The main process parameters.

Parameter	Laser Power (W)	Scanning Speed (mm/s)	Z-Axis Lift (mm)	Cooling Method	Powder Feeding Rate (RPM)
	1200	10	0.7	Air / Ice water	1.4

**Table 2 materials-14-03734-t002:** Numerical variation of power reduction step by step.

Layer	1	2	3	4	5	6	7	8	9	10	11	12	13	14
Laser power (W)	1200	1175	1150	1125	1100	1090	1080	1070	1060	1050	1045	1040	1035	1030

**Table 3 materials-14-03734-t003:** Tensile data of the yield strength, tensile strength, and elongation.

Condition	Yield Strength (MPa)	Tensile Strength (MPa)	Elongation (%)
Air	423.75 ± 19	723.03 ± 12	26.4 ± 2.1
Ice water	457.482 ± 22	759.64 ± 18	23.1 ± 3.1

## Data Availability

Data sharing not applicable.

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
