# Peer review of "Effect of Cooling Method on Formability of Laser Cladding IN718 Alloy"

_materials, 2021, doi:10.3390/ma14133734_

Round 1

Reviewer 1 Report

Dear authors,

I have read your manuscript with interests and attention. 
Generally, the subject sims to be interesting, however in present form I suggest significant rewriting the paper and resubmit your manuscript.

The number of imperfection is to high as for journal with so high IF.
Please consider my general and detailed comments, which could help you to improve you article:

The introduction of Inconel is weak, and rather related to other materials with different metallurgical properties.

Temperature modeling results should not be considered and described as absolute, but rather relative.

Line 80 - The sentence is not properly worded, although you can guess what it is about.

Line 173 - symbol of watt is W.

Line 174  - 1.4r/min I suggest to change on RPM.

Figures 6 and 7 are illegible.

Lines 322-323 - The sentence is very confusing.

Line 324 I think you are overusing the word "cladding", sometimes "layer" is fine.

There are many sentences in the study that are difficult to understand due to the poor structure of the sentences.

The condition of independent repeatability of the experiment is not met.
There is no unambiguous information about the geometry of the samples, there is no compliance in this respect in Figs. 7 and 9.

It was not specified where on the padding weld length the cross-section for metallographic tests was made.Measurements presented in tab. 2 are statistically unreliable.

The mean value and standard deviation should be reported

I suggest significant rewriting the paper and resubmit your manuscript.

Regards

Author Response

Thank you very much for your thorough reading of our manuscript and your valuable suggestions on improving our paper.

The corrections are listed below. Our specific responses with regard to the mandatory corrections are highlighted by purple color in the manuscript.

Reviewer 2 Report

In presentet manuscript the authors the finite element model of temperature field of straight thin-walled parts in laser cladding IN718 was established. Also  the growth of microstructure was simulated by cellular automata method through macro-micro coupling.

The authors statet that the results simulation results are in good agreement with the microstructure of the cladding layer observed by the experiment. They found that the used forced water cooling device in the process of laser cladding IN718 alloy, the cooling rate and solidification rate are improved, the primary dendrite arm spacing of columnar dendrite is reduced, the hardness of the cladding layer is improved, the yield strength and tensile strength are enhanced, and the elongation is reduced.

The presented manuscript of the authors are quite good. The some issue are require completion in manuscript. Therefore before publishing should be consider the following comments:

  1. The abstract is too general.
  2. Were there any cracks in the microstructure? Often, they can be appear during quite intensive cooling.
  3. Please present the experimental data in the table. Please add a subsection „Materials and methods”.
  4. What test equipment was used in this study? I mean the type of laser, kinde of microscope to observation microstructure, microhardness tester or strength test machine.
  5. How many repetitions were made for each sample?
  6. Please (if is possible) add the enlarged area of cooling device during the laser cladding on Figure 5.
  7. The left part of Figure 4 is not described.

Author Response

(The authors gave the same response as above.)

Reviewer 3 Report

The authors have studied the effect of the cooling method on the formability of laser clad IN718 alloy. The authors used finite element tools for establishing the microstructural properties of the cladding layer. This work done is comprehensive and may be of high interest to the materials research community for developing laser clads. materials. However, some important suggestions are for the authors as follows:

The authors should mention the number of samples used for tensile tests. There is not a big difference in the ultimate and yield tensile strength values. Of course, high elongation but less significant. The authors should justify these observations with recent literature.

Figures 24-26 can be clubbed together to make the flow easier. Similarly, if it is possible, Fig. 12-15 can be clubbed together.

Author Response

(The authors gave the same response as above.)

Round 2

Reviewer 1 Report

Dear Authors,

I have read your manuscript after revision with attention and interest.
The visible improvement has been noticed.

However, I have a few significant comments listed bellow:

  1. Line 32. "good weldability at low temperature" I suggest to extend this sentence so that the text has a chance to be understandable.
  2. Line 49. You have been writing "More epitaxial oriented structures were obtained" in my opinion this sentence sims to be incorrect because considered epitaxial growth cannot be graded, it is either there or not.
  3. The descriptions of fig. 7 and 8 are completely illegible.
  4. The table 3. Mpa shift please on MPa

Regards

Author Response

(The authors gave the same response as above.)

Reviewer 2 Report

The authors responded to my all comments. I recommend publishing this manuscript.

Author Response

Thank you very much for your thorough reading of our manuscript and your recognition of the paper.